# A Comprehensive Review of the Contribution of Mitochondrial DNA Mutations and Dysfunction in Polycystic Ovary Syndrome, Supported by Secondary Database Analysis

**DOI:** 10.3390/ijms26031172

**Published:** 2025-01-29

**Authors:** Hiroshi Kobayashi, Sho Matsubara, Chiharu Yoshimoto, Hiroshi Shigetomi, Shogo Imanaka

**Affiliations:** 1Department of Gynecology and Reproductive Medicine, Ms.Clinic MayOne, 871-1 Shijo-cho, Kashihara 634-0813, Japan; shogo_0723@naramed-u.ac.jp; 2Department of Obstetrics and Gynecology, Nara Medical University, 840 Shijo-cho, Kashihara 634-8522, Japan; s.matsubara@kei-oushin.jp (S.M.); chiharu-y@naramed-u.ac.jp (C.Y.); hshige35@gmail.com (H.S.); 3Department of Medicine, Kei Oushin Clinic, 5-2-6 Naruo-cho, Nishinomiya 663-8184, Japan; 4Department of Obstetrics and Gynecology, Nara Prefecture General Medical Center, 2-897-5 Shichijyonishi-machi, Nara 630-8581, Japan; 5Department of Gynecology and Reproductive Medicine, Aska Ladies Clinic, 3-3-17 Kitatomigaoka-cho, Nara 634-0001, Japan

**Keywords:** mitochondrial DNA copy number, mitochondrial DNA mutations, oxidative stress, polycystic ovary syndrome, replication errors

## Abstract

Polycystic ovary syndrome (PCOS) is a common endocrine disorder affecting women of reproductive age characterized by a spectrum of clinical, metabolic, reproductive, and psychological abnormalities. This syndrome is associated with significant long-term health risks, necessitating elucidation of its pathophysiology, early diagnosis, and comprehensive management strategies. Several contributory factors in PCOS, including androgen excess and insulin resistance, collectively enhance oxidative stress, which subsequently leads to mitochondrial dysfunction. However, the precise mechanisms through which oxidative stress induces mitochondrial dysfunction remain incompletely understood. Comprehensive searches of electronic databases were conducted to identify relevant studies published up to 30 September 2024. Mitochondria, the primary sites of reactive oxygen species (ROS) generation, play critical roles in energy metabolism and cellular homeostasis. Oxidative stress can inflict damage on components, including lipids, proteins, and DNA. Damage to mitochondrial DNA (mtDNA), which lacks efficient repair mechanisms, may result in mutations that impair mitochondrial function. Dysfunctional mitochondrial activity further amplifies ROS production, thereby perpetuating oxidative stress. These disruptions are implicated in the complications associated with the syndrome. Advances in genetic analysis technologies, including next-generation sequencing, have identified point mutations and deletions in mtDNA, drawing significant attention to their association with oxidative stress. Emerging data from mtDNA mutation analyses challenge conventional paradigms and provide new insights into the role of oxidative stress in mitochondrial dysfunction. We are rethinking the pathogenesis of PCOS based on these database analyses. In conclusion, this review explores the intricate relationship between oxidative stress, mtDNA mutations, and mitochondrial dysfunction, offers an updated perspective on the pathophysiology of PCOS, and outlines directions for future research.

## 1. Introduction

Polycystic ovary syndrome (PCOS) is a prevalent endocrine disorder affecting women of reproductive age characterized by oligo-ovulation or anovulation, hyperandrogenism, and polycystic ovarian morphology, alongside metabolic disturbances, such as insulin resistance, obesity, dyslipidemia, and an increased risk of type 2 diabetes [1,2,3,4,5]. The condition also leads to reproductive challenges and psychological disorders. The Rotterdam criteria, widely used for diagnosing PCOS, require the presence of at least two of three diagnostic features: oligo- or anovulation, clinical or biochemical hyperandrogenism, and ultrasonographic evidence of polycystic ovarian morphology [6,7]. In contrast, the diagnostic criteria established by the Japan Society of Obstetrics and Gynecology in 2024 adopt a more stringent approach, necessitating the fulfillment of all three conditions: (1) irregular menstruation, (2) polycystic ovarian morphology or elevated anti-Müllerian hormone (AMH) levels, and (3) hyperandrogenism or elevated luteinizing hormone (LH) levels [8]. PCOS is recognized as multifactorial, with genetic, environmental, and transgenerational factors contributing to its development [5,6,9,10]. The pathophysiology primarily involves two mechanisms: excessive androgen production and insulin resistance, which independently or interactively contribute to the clinical manifestations, including reproductive, dermatologic, and metabolic features. Hyperandrogenism is central to the disorder, often resulting from dysregulation of the hypothalamic–pituitary–ovarian axis or ovarian dysfunction [11,12,13,14,15]. Additionally, insulin resistance, a hallmark feature of PCOS, substantially contributes to its metabolic and reproductive manifestations and is often accompanied by compensatory hyperinsulinemia [16]. Extensive studies have examined the molecular interplay between androgen excess and insulin resistance [17].

Emerging evidence highlights mitochondrial dysfunction as a key factor in PCOS [18]. Studies indicate compromised mitochondrial function and morphology in ovarian tissues, which affect metabolic and reproductive processes [9,19]. Androgen excess disrupts mitochondrial electron transport, depletes ATP, and impairs mitochondrial biogenesis and mitophagy [19,20]. Additionally, insulin resistance exacerbates mitochondrial dysfunction by altering metabolism and increasing oxidative stress [5]. This cycle of oxidative stress and mitochondrial dysfunction contributes to the progression of PCOS. To investigate mitochondrial dysfunction in PCOS, researchers have explored mutations in mitochondrial DNA (mtDNA) and variations in its copy number [21]. Advances in genetic analysis techniques, including next-generation sequencing, have identified point mutations and deletions in mtDNA potentially associated with PCOS. While substantial evidence indicates that mitochondrial dysfunction driven by oxidative stress plays a pivotal role in the pathogenesis of PCOS, investigations into mtDNA mutations have not yet yielded conclusive evidence supporting this link. Consequently, we undertook an effort to compile and analyze the existing data on mtDNA mutations reported in connection with PCOS. Understanding the factors contributing to mtDNA mutations and deletions, as well as their repair mechanisms, is essential for elucidating the pathophysiology of PCOS and translating these findings into therapeutic strategies.

This review consolidates current knowledge on oxidative stress and mitochondrial dysfunction in PCOS, with a particular focus on mtDNA damage and repair mechanisms, and proposes future directions for advancing the understanding of its pathogenesis.

## 2. The Impact of Androgen Excess and Insulin Resistance on Mitochondrial Function

Here, we provide an analysis of the impact of androgen excess and insulin resistance on mitochondrial function and oxidative stress, underscoring their pivotal roles in the pathophysiology of this disorder.

### 2.1. Androgen Excess

Androgens in the human ovary are not only precursors for estrogen synthesis but al-so directly influence follicular development, steroidogenesis, and ovarian reserve maintenance. Their precise levels are critical for normal ovarian function. Androgens regulate fatty acid metabolism via the androgen receptor, expressed in tissues like adipose, the liver, skeletal muscle, and the gonads [22]. They activate adipogenic pathways, amplify lipotoxicity, modulate adipokine secretion, regulate mitochondrial biogenesis, support anabolic processes, and induce pro-inflammatory cytokines. Additionally, androgens influence insulin signaling, a key regulator of adipogenesis, particularly in visceral adipose tissue. Elevated androgen levels, as seen in PCOS, contribute to central obesity and metabolic dysfunction. This review explores five mechanisms linking androgens to PCOS pathophysiology. Firstly, androgens activate adipogenic pathways by modulating adipocyte differentiation, lipid metabolism, and preadipocyte function, contributing to PCOS-related dysfunctions (Figure 1). They upregulate fatty acid synthase (FAS) [23] and acetyl-CoA carboxylase (ACC) [24], driving lipid biosynthesis while inhibiting carnitine palmitoyltransferase 1 (CPT1), reducing fatty acid oxidation and energy production [25] (Figure 1, Lipogenic pathways). Secondly, androgens exacerbate lipotoxicity by promoting fatty acid accumulation in non-adipose tissues [26], leading to insulin resistance—a PCOS hallmark (Figure 1, Lipotoxicity). Elevated androgens enhance lipolysis, increasing free fatty acids (FFAs), which impair mitochondrial function, disrupt oxidative phosphorylation, and induce lipotoxicity, further impairing insulin signaling [27,28]. Thirdly, androgens alter adipokine secretion, reducing adiponectin and increasing leptin, contributing to metabolic disturbances like insulin resistance in PCOS and related conditions [29,30] (Figure 1, adipokine). Fourthly, androgen excess impairs mitochondrial biogenesis and dynamics, disrupting energy metabolism, insulin resistance, and ovarian function [19,26] (Figure 1, Anabolism). By dysregulating pathways like PGC-1α [31], AMPK [32], and mTOR [33], androgens reduce mitochondrial quality and quantity, worsening metabolic dysfunctions [34]. Fifthly, androgens promote anabolic processes through the PI3K/AKT and mTOR pathways, enhancing glucose metabolism to support energy-intensive activities [35] (Figure 1, Anabolism). However, excessive levels impair insulin signaling [36], triggering hyperglycemia, dyslipidemia, and T2DM risks, emphasizing tissue-specific and dose-dependent effects. Lastly, androgens induce chronic low-grade inflammation, increasing pro-inflammatory cytokines like TNF-α and IL-6 and activating stress pathways that impair insulin signaling [37] (Figure 1, Cytokine). This inflammatory cascade exacerbates insulin resistance, ovarian dysfunction, and cardiovascular risks in PCOS. Dysregulation of lipid and glucose metabolism, inflammation, and mitochondrial dynamics under androgen excess contributes to cellular energy imbalance and PCOS pathophysiology.

Subsequently, we elucidate the mechanisms through which androgens stimulate the pro-duction of reactive oxygen species (ROS). Androgens stimulate the production of reactive oxygen species (ROS) through several mechanisms. First, an imbalance between oxidative capacity and fatty acid supply reduces mitochondrial respiration efficiency, leading to ROS generation, particularly superoxide anions [38]. Hyperandrogenism suppresses antioxidant enzymes like superoxide dismutase (SOD) and glutathione peroxidase (GPx), impairing ROS neutralization and exacerbating mitochondrial damage [39]. Second, androgens activate NADPH oxidase (NOX), a major source of ROS, which contributes to insulin resistance, endothelial dysfunction, and inflammation [40]. NOX activation also promotes pro-inflammatory cytokines like TNF-α and IL-6 [22,41]. Third, androgen-induced alterations in mitochondrial dynamics, including disrupted fission and fusion, impair the electron transport chain (ETC) and enhance ROS production [42]. Fourth, androgen-modulated endoplasmic reticulum (ER) stress activates the unfolded protein response (UPR), increasing ROS and inflammatory signaling [43]. Finally, oxidative stress upregulates cytochrome P450 17A1 (CYP17A1), enhancing androgen production [22,44].

Taken together, excess androgens affect tissue metabolism, promote lipolysis, and redistribute fat, especially in visceral depots, while disrupting insulin signaling and lipid metabolism. These effects impair mitochondrial function by disrupting bioenergetics, activating NOX, and altering mitochondrial dynamics, resulting in increased ROS production. Mitochondrial dysfunction exacerbates oxidative stress, creating a cycle of cellular injury [22,45]. Elevated androgen levels also impair ovarian mitochondrial function, leading to abnormal follicular development, compromised ovulation, and reduced reproductive outcomes [46]. Excessive oxidative stress and mitochondrial dysfunction are central to the pathogenesis of PCOS.

### 2.2. Insulin Resistance

Insulin, a key metabolic regulator, enhances glycolysis, glycogenesis, lipogenesis, and protein synthesis while suppressing glycogenolysis, gluconeogenesis, lipolysis, and protein degradation [47,48,49]. During high energy availability, insulin downregulates PGC-1α transcription via pathways like the PI3K–AKT–FOXO axis, reducing reliance on oxidative metabolism [50,51] (Figure 1, mitochondrial function). Insulin inhibits gluconeogenesis by suppressing phosphoenolpyruvate carboxykinase (PEPCK) and glucose-6-phosphatase (G6Pase) transcription through PI3K-AKT signaling and FOXO1 modulation [52] (Figure 1, gluconeogenesis). It also promotes glucose uptake by facilitating glucose transporter 4 (GLUT4) translocation in insulin-sensitive tissues like skeletal muscle and adipose tissue [53] (Figure 1, glycolysis). Absorbed glucose undergoes glycolysis, fueling the TCA cycle and providing reducing equivalents (nicotinamide adenine dinucleotide (NADH), flavin adenine dinucleotide (FADH2)) for the ETC, enhancing ATP production via OXPHOS [54]. Insulin regulates lipid metabolism by activating ACC to promote fatty acid synthesis while inhibiting lipolysis, reducing FFA release [55] (Figure 1, lipogenic pathways). This prevents mitochondrial overload, preserves redox homeostasis, and supports mitochondrial function. Insulin enhances ETC efficiency to minimize ROS formation and upregulates antioxidant enzymes like SOD and GPx to mitigate oxidative stress [56].

PCOS is closely linked to insulin resistance, driving metabolic and reproductive dysfunctions [22,56]. Insulin resistance exacerbates mitochondrial dysfunction and suppresses PGC-1α expression [57]. Hyperinsulinemia triggers excessive androgen secretion, intensifying insulin resistance and hyperandrogenism in a pathological feedback loop [58,59]. Chronic hyperinsulinemia promotes fatty acid accumulation through increased lipogenesis and impaired lipolysis. Excess FFAs transported to the liver and skeletal muscle overwhelm mitochondrial oxidative capacity, leading to incomplete oxidation and O2•− production [60]. Insulin activates glycolysis by enhancing phosphofructokinase-1 (PFK1) and liver-type pyruvate kinase (LPK) activities, favoring carbohydrate metabolism over fatty acid oxidation [61]. Excessive acetyl-CoA strains the ETC, increasing ROS production. Elevated FFAs also induce ER stress, activating stress kinases like JNK and PKR-like ER kinase (PERK), further contributing to ROS generation [62,63,64,65]. Chronic hyperinsulinemia drives low-grade inflammation, releasing cytokines like TNF-α that activate NOX and other ROS-producing pathways [66].

Collectively, insulin resistance promotes ROS generation via hyperglycemia, elevated FFAs, impaired fatty acid oxidation, chronic inflammation, NOX activation, and ER stress [22]. While insulin regulates glucose and lipid homeostasis, insulin resistance and hyperinsulinemia amplify oxidative stress, contributing to diseases like T2DM and cardiovascular disorders [56,58,59]. Oxidative stress biomarkers like 8-oxo-dG are elevated in the follicular fluid and serum of women with PCOS, disrupting follicular development and fertility [22,67,68]. Experimental models mimicking PCOS showed significant oxidative stress and mitochondrial damage in the pregnant uterus [69]. ROS, such as superoxide radicals and hydrogen peroxide, induce DNA damage [70], increasing risks for endometrial, ovarian, and breast cancers in women with PCOS [71]. The interplay between androgen excess [6] and insulin resistance [56] perpetuates oxidative stress and mitochondrial dysfunction, which are central to PCOS pathophysiology.

## 3. The Impact of Oxidative Stress on the Pathophysiology of PCOS

Oxidative stress plays a critical role in reproductive disorders, including PCOS, endometriosis, unexplained infertility, and pregnancy complications like preeclampsia [72]. Numerous studies highlight that reducing oxidative stress can alleviate PCOS symptoms. While animal and in vitro study findings vary, antioxidant supplementation shows promise in lowering ROS levels [72]. Ding et al. [73] studied 30 female Sprague-Dawley rats divided into control (*n* = 10), PCOS-IR (*n* = 10), and MitoQ10 treatment (*n* = 10) groups to evaluate MitoQ10, a mitochondria-targeted antioxidant. MitoQ10, a modified coenzyme Q10 derivative, protects mitochondria from oxidative damage. Results showed MitoQ10 reduced pro-apoptotic proteins (cytochrome c, Bcl-2-associated X) and increased anti-apoptotic protein Bcl-extra large, improving insulin resistance and mitochondrial dysfunction [73]. Yang et al. [74] examined metformin’s effects on mtDNA copy number in PCOS patients. Metformin, a hypoglycemic agent for T2DM, also exhibits antioxidant properties, making it a potential therapy for oxidative-stress-linked conditions like PCOS and cardiovascular diseases. After one year of metformin treatment, mtDNA copy numbers gradually declined, suggesting improved mitochondrial function [74]. Antioxidant therapies targeting oxidative stress and mitochondrial dysfunction may enhance PCOS outcomes, but the protective effects of MitoQ10 and metformin against mtDNA mutations require further study [73]. For a detailed review of antioxidant efficacy in PCOS, see Iervolino et al. [75].

## 4. Mechanisms for Maintaining Mitochondrial Genome Stability

The intricate relationship between mitochondrial function and mtDNA mutations is significant. It is plausible to posit that mtDNA mutations impair mitochondrial function, while mitochondrial dysfunction further aggravates these mutations. Such mutations impair the OXPHOS system, reducing energy efficiency and increasing oxidative stress, with effects depending on mutation type, heteroplasmy level, and tissue energy demands. These mutations are central to inherited mitochondrial diseases and acquired dysfunctions from aging and chronic conditions. Understanding mtDNA structure and function is key to studying mutation and repair mechanisms (Figure 2). The mitochondrial genome is a 16,569-bp, circular, double-stranded molecule encoding 13 OXPHOS proteins, two rRNAs, and 22 tRNAs [76]. It includes the heavy (H) and light (L) strands, with the H strand encoding most proteins. mtDNA has two regions: the D-loop and the non-D-loop. The D-loop (16,025–576 bp) is a non-coding regulatory region crucial for replication and transcription, containing OriH and OriL origins and promoter sequences [77,78]. OriH initiates H-strand replication, while OriL starts L-strand replication, ensuring mtDNA maintenance [77]. D-loop mutations disrupt replication and transcription, causing mitochondrial dysfunction [78]. The non-D-loop region (577–16,024 bp) encodes genes for mitochondrial bioenergetics and metabolism; mutations impair protein synthesis and OXPHOS, leading to dysfunction and disease. While mtDNA encodes 13 OXPHOS proteins, other required proteins are encoded by nuclear genes [79]. Mitochondrial function relies on mitochondrial and nuclear genome interplay [78]. Unlike the nuclear genome, mtDNA has multiple copies per cell (10–10,000, up to 200,000 in oocytes) [80]. This redundancy highlights the importance of mtDNA stability for mitochondrial function.

Evolution has equipped mitochondria with mechanisms to maintain genomic stability, including the mitochondrial bottleneck theory, fusion and fission dynamics, mitophagy, and mtDNA repair pathways like base excision repair (BER) (Figure 3). The mitochondrial bottleneck theory explains changes in mtDNA composition across generations despite limited maternal inheritance [81]. During early oogenesis, mtDNA copy numbers are dramatically reduced in primordial germ cells, forming a “bottleneck” that limits the mtDNA population. This reduction and subsequent amplification during oocyte development result in random sampling, causing variability in heteroplasmy between offspring and mothers. Mitochondria aim to preserve their genomes with minimal mutations. Through fusion and fission, mitochondria with mutations merge with healthy ones, enabling functional complementation and reducing defects [19]. Damaged mitochondria are removed via mitophagy, a process critical for mitochondrial quality control and maintaining cellular integrity [19]. Mitophagy ensures energy production and prevents mitochondrial dysfunction linked to aging, neurodegenerative diseases, and cancer. Repairing damaged mtDNA and maintaining replication machinery integrity are key to genomic stability. However, mitochondrial repair systems (e.g., BER) lack the fidelity of nuclear DNA repair, leading to errors. Mitochondrial repair and replication systems address small base lesions from oxidative stress, deamination, or hydrolysis [82,83]. Key enzymes in this pathway include Uracil-DNA glycosylase (UNG), Thymine-DNA glycosylase (TDG), AP Endonuclease (APE1), DNA polymerase γ, DNA polymerase β, and DNA ligase [83]. Despite its importance, BER is prone to errors during base removal, gap filling, or ligation. Consequently, mtDNA replication is less regulated than nuclear DNA replication. Errors also occur during mtDNA duplication when DNA polymerase incorporates incorrect bases or mismanages structural elements [5,84]. High mtDNA replication and transcription rates, driven by energy demands, increase mutation risks and structural rearrangements, which can contribute to cancer, neurodegenerative disorders, and aging-related diseases.

## 5. Types of mtDNA Mutations Associated with Oxidative Stress

Genomic mutations generally involve base substitutions, which can be categorized into transition and transversion mutations [85]. Transition mutations replace a purine (adenine or guanine) with another purine or a pyrimidine (cytosine or thymine) with another pyrimidine. These mutations are among the most prevalent forms of single-nucleotide polymorphisms (SNPs) in DNA. Causes of transition mutations include tautomerization, deamination, replication errors, and CpG hotspots [86]. Tautomeric shifts can lead to incorrect DNA pairing, and deamination converts cytosine to uracil, resulting in a C→T transition. Adenine deaminates to hypoxanthine, mispairing with cytosine, causing an A→G transition. Guanine deaminates to xanthine, which may cause replication errors. Although repair mechanisms correct these errors, they are not always fully effective. In mtDNA, transition mutations are more frequent in the heavy (H) strand [87] due to its prolonged single-stranded state during replication, making it more vulnerable to mutational processes like deamination. These mutations often result from replication errors by mtDNA polymerase gamma or spontaneous deamination. The light (L) strand, being cytosine-rich, is more susceptible to oxidative damage due to increased exposure during replication [88]. Transversion mutations, which involve the substitution of a purine with a pyrimidine (e.g., A ↔ C, G ↔ T), are less common. ROS can oxidize DNA bases, leading to transversion mutations. For example, oxidative stress can produce 8-oxo-dG, which mispairs with adenine, resulting in G:C to T:A transversions. These mutations highlight the impact of oxidative stress on mitochondrial genomic stability.

A 2014 Drosophila study examined the frequency and distribution of mtDNA mutations with age [89]. Most mutations were transition mutations, while transversions, indicative of oxidative damage, were rare. The loss of mitochondrial Sod2 function or the DNA repair enzyme Ogg1 did not affect mtDNA mutation frequency. Kennedy et al. [90] observed that older brain samples had more mutations, but transition mutations occurred more frequently than transversions. The frequency of oxidative-stress-dependent transversions, like G→T and C→A, was low and did not increase with age in human brain samples [91]. The most common transition mutations, G→A or C→T, result from base misincorporation or cytosine deamination by DNA polymerase γ [90]. The second most frequent transitions, T→C or A→G, arise from adenosine deamination or T-dGTP mispairing by DNA polymerase γ [90]. Mutation burden increases with age across species, with proofreading-deficient mice showing a higher point mutation burden [91]. However, translocation mutations remained unchanged with age, suggesting oxidative damage is not the main driver of mtDNA mutations [91]. Studies indicate that mtDNA mutations are primarily due to replication errors, such as misincorporation by DNA polymerase γ or deamination of cytidine and adenosine, rather than oxidative stress [89,90,91,92]. Similarly, Matkarimov et al. argue that if oxidative stress were the primary driver, transversion mutations would be more prevalent [93].

## 6. An Overview of the Principal Types of mtDNA Mutations Associated with PCOS

Due to its proximity to the ETC within the inner mitochondrial membrane and the lack of protective histones, mtDNA is highly susceptible to oxidative damage from ROS, leading to a significantly higher mutation rate than nuclear DNA [89,94,95,96]. One potential cause of mtDNA mutations is oxidative stress from androgen excess or insulin resistance (see Section 3). This raises the following question: does oxidative stress in PCOS patients directly induce mtDNA damage? This section reviews the types of mtDNA mutations in PCOS patients, including point mutations, deletions, polymorphisms, and heteroplasmy. Mitochondrial point mutations can have substantial phenotypic effects depending on the mutation type, location, and proportion of mutated mtDNA (heteroplasmy). Minor mutations generally do not result in severe disease manifestations, with most healthy individuals carrying less than 1% point mutations without noticeable effects [95]. Polymorphisms, common genetic variations present in over 1% of the population, are generally neutral and contribute to genetic diversity. In PCOS, mtDNA polymorphisms affect pathways like androgen biosynthesis [97], insulin resistance [98], and ovarian function [99]. Variants in genes, such as CYP17A1 [100], LHR [101], FSHR [102], IRS1 [103], TNF-α [104], and IL-6 [105], are linked to PCOS features like hyperandrogenism, anovulation, and metabolic disturbances. Elevated mtDNA mutations and heteroplasmy have been observed in PCOS patients [77,84,106], with higher incidence in mt-tRNA and mt-rRNA variants compared to the general population [84]. High heteroplasmy levels may impair mitochondrial function in ovarian cells, contributing to PCOS development, although clinical symptoms appear only when the mutated mtDNA exceeds a critical threshold, known as the threshold effect. Therefore, mutations in heteroplasmic cases do not always lead to disease. A 2024 review by Zhou et al. discussed mitochondrial heterogeneity in ovarian diseases and emerging therapeutic strategies [95].

Below is a summary of the main mtDNA mutations in PCOS.

### 6.1. Point Mutations

Point mutations involve single nucleotide changes within the mtDNA sequence. Mitochondrial respiratory chain complexes I (MT-ND1, MT-ND2, MT-ND3, MT-ND4, MT-ND4L, MT-ND5, MT-ND6), III (MT-CYB), IV (MT-COX1, MT-COX2, MT-COX3), and V (MT-ATP6, MT-ATP8) consist of both mitochondrial-encoded and nuclear-encoded genes (Figure 2). Mutations were observed in MT-ND1, ND2, and ND5 [21,106], as well as in Mt-tRNA and Mt-rRNA [21,48,107,108,109,110,111]. Frameshift mutations were predominantly seen in MT-ND2 and ND5 genes [112]. Most pathogenic mutations were in the MT-CO3 gene, with additional mutations in NC5 [21,84]. Heteroplasmic mutations in Complex I and IV subunit genes may impair electron transfer, reducing ATP synthesis and increasing ROS production [84]. These mtDNA mutations can disrupt electron transport, induce energy deficits, and increase ROS, leading to mitochondrial dysfunction.

Next, we summarize mtDNA mutations in PCOS patients. In 2020, next-generation sequencing of mtDNA in 30 PCOS women revealed 67 mutations in the D-loop and OriL regions, with 82% identified as transition substitutions [77]. Bibi et al. [112] found that the D-loop region had the highest mutation frequency. In 2021, Dabravolski et al. [107] cataloged mtDNA mutations in PCOS individuals, revealing that 29 out of 35 mutations (83%) were transitions and 6 (17%) were transversions. Data from previous studies were used to recalculate transition and transversion mutation proportions [5,48,77,84,94,106,109,110,112,113,114]. Of 225 documented mutations, 199 (88.4%) were transitions, and 26 (11.6%) were transversions (Appendix A). Among 111 D-loop mutations, 98 (88.3%) were transitions, and 106 (89.8%) of 118 non-D-loop mutations were also transitions (Appendix A). No significant differences in mutation patterns were observed between D-loop and non-D-loop regions. In conclusion, transition mutations are the most frequent mtDNA alterations in PCOS patients.

### 6.2. Deletions

The 4977 bp deletion, known as the “common deletion”, results in a loss of nearly one-third of the mitochondrial genome [115]. This deletion includes five tRNA genes and seven genes encoding subunits of mitochondrial complexes I, IV, and V, which are essential for mitochondrial energy production [80]. The deletion arises due to replication slippage, where repetitive sequences cause polymerase misalignment during replication [116]. The 4977 bp deletion has been linked to metabolic disorders like T2DM [117], aging [118], and ovarian aging [119], suggesting its involvement in both metabolic and reproductive abnormalities. Although the deletion is not exclusive to PCOS, emerging studies propose its potential involvement in the pathophysiology of the syndrome [80]. Thus, the 4977 bp deletion likely results from replication slippage and accumulates with age.

### 6.3. Mutations in tRNA Genes

Nine distinct mt-tRNA mutations have been identified in PCOS patients with insulin resistance, including mutations in mt-tRNA^Leu(UUR)^ A3302G, C3275A, mt-tRNA^Gln^ T4363C, T4395C, mt-tRNA^Ser(UCN)^ C7492T, mt-tRNA^Asp^ A7543G, mt-tRNA^Lys^ A8343G, mt-tRNA^Arg^ T10454C, and mt-tRNA^Glu^ A14693G [110] (Appendix A). Mutations in mt-tRNA disrupt its tertiary structure, impair aminoacylation, and affect mitochondrial RNA functionality [114]. Individuals with specific mt-tRNA mutations show impaired mt-tRNA metabolism, reduced mitochondrial membrane potential, decreased ATP production, disrupted protein synthesis, and increased ROS levels, possibly contributing to pancreatic β-cell dysfunction and apoptosis, leading to T2DM [78,114].

#### 6.3.1. tRNA^Lys^

Mutations in the MT-TK gene, encoding mt-tRNA^Lys^, have been found in PCOS patients [110]. This mutation is associated with myoclonic epilepsy with ragged-red fibers (MERRF) syndrome, a rare hereditary disorder [120]. MERRF impairs mitochondrial protein synthesis, leading to dysfunctional OXPHOS and symptoms like myoclonic epilepsy, muscle weakness, and ragged-red fibers. MERRF also correlates with endocrine abnormalities, such as T2DM, thyroid dysfunction, and, in some cases, hyperandrogenism or menstrual irregularities, linking mitochondrial dysfunction in both MERRF and PCOS.

#### 6.3.2. tRNA^Leu(UUR)^

The tRNA ^Leu(UUR)^ gene, which decodes leucine, is linked to several mitochondrial disorders when mutated, as it disrupts mitochondrial protein synthesis and energy production. Commonly associated conditions include MELAS syndrome, T2DM, deafness, cardiomyopathy, and myopathy [121]. A study of a Chinese family showed a connection between a homoplasmic A3302G mutation in the tRNA ^Leu(UUR)^ gene and T2DM with insulin resistance in earlier generations and PCOS in a third-generation member, suggesting a potential link between this mutation and PCOS pathogenesis [122].

#### 6.3.3. tRNA^Ser(UCN)^

Mutations in the MT-TS1 gene, which encodes tRNA^Ser(UCN)^, disrupt mitochondrial protein synthesis, leading to disorders like mitochondrial myopathy, hearing loss, PEO, chronic neuropathy, and Leigh syndrome [123]. In Chinese PCOS patients, homoplasmic ND5 T12338C and tRNA^Ser(UCN)^ C7492T mutations were identified, although their clinical significance remains uncertain [21,110].

### 6.4. Types of mtDNA Mutations in Diseases Beyond PCOS

Oxidative stress contributes to the progression of age-related degenerative diseases and cancer. mtDNA mutations were studied in normal colon tissue, ulcerative colitis, and colorectal cancer, revealing an increase in mutations during early dysplasia, which diminished as cancer progressed [124]. Most mutations were C→T transitions on the heavy strand caused by replication errors. A similar pattern of transition mutations was observed in endometrial cancer [125], glaucoma [126], and age-related macular degeneration [127], suggesting that oxidative-stress-related mtDNA mutations are characterized by transition substitutions, as seen in PCOS.

## 7. mtDNA Copy Number Variations in PCOS

This section analyzes mtDNA copy number variations in PCOS patients, a key marker of mitochondrial content and functionality, with fluctuations linked to various chronic diseases [80,128]. Recent studies have highlighted inconsistent findings in mtDNA copy numbers in PCOS [80]. Data compiled from previously published studies were systematically analyzed to assess variations in mtDNA copy number (Table 1). A review of ten studies showed eight reporting a decrease in mtDNA copy number [5,77,84,106,110,113,114,129,130] and two showing an increase [80,106]. These variations may relate to the severity of PCOS symptoms [129]. Reduced mtDNA copy numbers in PCOS suggest that oxidative stress, insulin resistance, hormonal dysregulation, inflammation, and mitochondrial dysfunction hinder mtDNA replication. Conversely, some cells may increase mtDNA copy numbers to compensate for mitochondrial damage and meet metabolic demands. Consistent with the findings of Ye et al. [80], a study with induced pluripotent stem cells from PCOS patients confirmed an increase in mtDNA copy numbers [131], possibly as a compensatory response [78]. In conclusion, mtDNA copy number alterations in PCOS highlight the importance of studying both quantitative and qualitative mtDNA abnormalities, offering insights into mitochondrial dynamics in PCOS.

## 8. Discussion

Mitochondrial dysfunction has gained attention in PCOS pathophysiology, mainly due to oxidative stress from androgen excess and insulin resistance. mtDNA is especially vulnerable to ROS damage due to its proximity to the ETC, lack of histones, and limited repair capacity [89,94,95,96]. PCOS-like symptoms have been observed in mitochondrial disorders with pathogenic mtDNA mutations, suggesting their potential role in PCOS etiology [9]. A 2023 review by Dong et al. found that mtDNA mutations drive mitochondrial dysfunction and contribute to PCOS-related insulin resistance through oxidative stress, energy deficits, and hormonal dysregulation [48]. Transition mutations are more frequent than transversions in PCOS and other oxidative-stress-related conditions [93,132]. The rarity of transversion mutations suggests that mtDNA mutations may result more from replication errors than oxidative damage [77,84,89,93,107]. Shukla et al. proposed that DNA polymerase γ replication errors, not ROS damage, primarily cause mtDNA mutations [77,84]. While mtDNA mutations are more common in PCOS patients, they do not always lead to disease manifestation. Most mutations in PCOS women are rare (<1% frequency), with few showing a minimum allele frequency (MAF) of 5% or higher [94]. Unlike nuclear DNA mutations, mtDNA mutations show more variability due to heteroplasmic dynamics, with factors like mtDNA copy number and mutation proportion being more critical in determining pathology. Mitochondria use mechanisms like the mitochondrial bottleneck theory, dynamics, and mitophagy to maintain genomic stability. It remains unclear whether mutated mtDNA causes mitochondrial dysfunction or is merely a by-product of PCOS. While oxidative stress is implicated in PCOS-related mitochondrial dysfunction [107], direct links between oxidative stress and mtDNA mutations are also elusive. Recent studies suggest that mtDNA somatic mutations accumulate with age in an organ-specific manner, with transversion mutations being rapidly cleared [133]. Understanding the functional significance of mtDNA mutations is essential for a deeper understanding of mitochondrial dysfunction in PCOS. Future studies should focus on the types of mtDNA mutations and their functional consequences. In conclusion, oxidative stress in PCOS is linked to mitochondrial dysfunction driven by androgen excess and insulin resistance, but its direct role in inducing dysfunction via mtDNA mutations remains unclear.

## 9. Future Prospects

Further investigation into mtDNA mutations in PCOS patients is essential to clarify their role, alongside oxidative stress, in ovarian dysfunction. Insights from mtDNA mutations could advance diagnosis and prognosis, but challenges remain for their clinical application. Genetic and environmental factors contribute to variability, complicating the identification of consistent biomarkers across diverse PCOS populations. The exact relationship between specific mtDNA mutations and PCOS development is not well-understood. Heteroplasmy, where mutated and wild-type mtDNA coexist, complicates evaluating mutation pathogenicity and its cellular impact. Many mutations may be passenger mutations, and determining the threshold for pathogenicity is challenging. Accurate detection of low-frequency mtDNA mutations in clinical samples requires highly sensitive methodologies [84]. Non-invasive diagnostic approaches, such as detecting mtDNA mutations in circulating mtDNA or follicular fluids, show promise. Advances in next-generation sequencing could improve the identification of mtDNA mutations as biomarkers for early detection, disease progression, and personalized therapies. However, mtDNA lacks established repair pathways and cannot be directly edited with tools like CRISPR-Cas9. Innovations in mitochondrial gene-editing techniques, like mitoTALENs and zinc-finger nucleases, offer potential for correcting mtDNA mutations [134]. Mitochondrial replacement therapy is another option, although it faces ethical and technical challenges [135]. Further research into mtDNA mutations in the ETC, metabolic reprogramming, and disease progression could reveal novel diagnostic and therapeutic targets. Addressing these challenges through interdisciplinary research could improve PCOS management and patient outcomes.

## Figures and Tables

**Figure 1 ijms-26-01172-f001:**
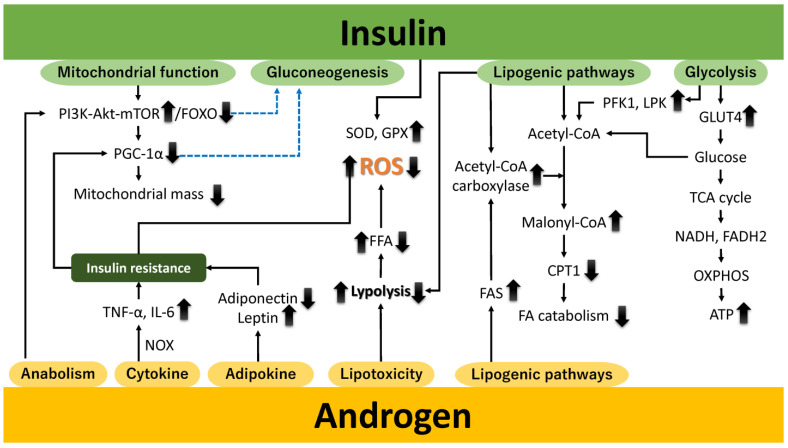
The effect of androgens (or androgen excess) and insulin (or insulin resistance) on physiological activity and oxidative stress. Elevated androgen levels can disrupt hormonal balance, alter metabolic processes, and promote cellular dysfunction. Insulin resistance impairs glucose uptake and metabolism, contributing to hyperinsulinemia and metabolic disturbances. Both factors synergistically enhance oxidative stress by increasing ROS production, impairing antioxidant defenses, and promoting mitochondrial dysfunction, ultimately leading to cellular damage and the exacerbation of pathophysiological conditions. Upward and downward arrows denote increases and decreases, respectively. The dashed blue line represents inhibition. Akt, protein kinase B; ATP, adenosine triphosphate; CPT1, carnitine palmitoyltransferase 1; FA, fatty acid; FADH2, flavin adenine dinucleotide; FAS, fatty acid synthase; FFA, free fatty acid; FOXO, Forkhead box O; GLUT4, glucose transporter 4; GPX, glutathione peroxidase; IL-6, interleukin nn-6; LPK, liver-type pyruvate kinase; mTOR, mechanistic target of rapamycin; NADH, nicotinamide adenine dinucleotide; NOX, nicotinamide adenine dinucleotide phosphate oxidase; OXPHOS, oxidative phosphorylation; PFK1, phosphofructokinase-1; PGC-1α, peroxisome proliferator-activated receptor gamma coactivator 1 alpha; PI3K, phosphatidylinositol 3-kinase; ROS, reactive oxygen species; SOD, superoxide dismutase; TCA, tricarboxylic acid; and TNF-α, tumor necrosis factor-alpha.

**Figure 2 ijms-26-01172-f002:**
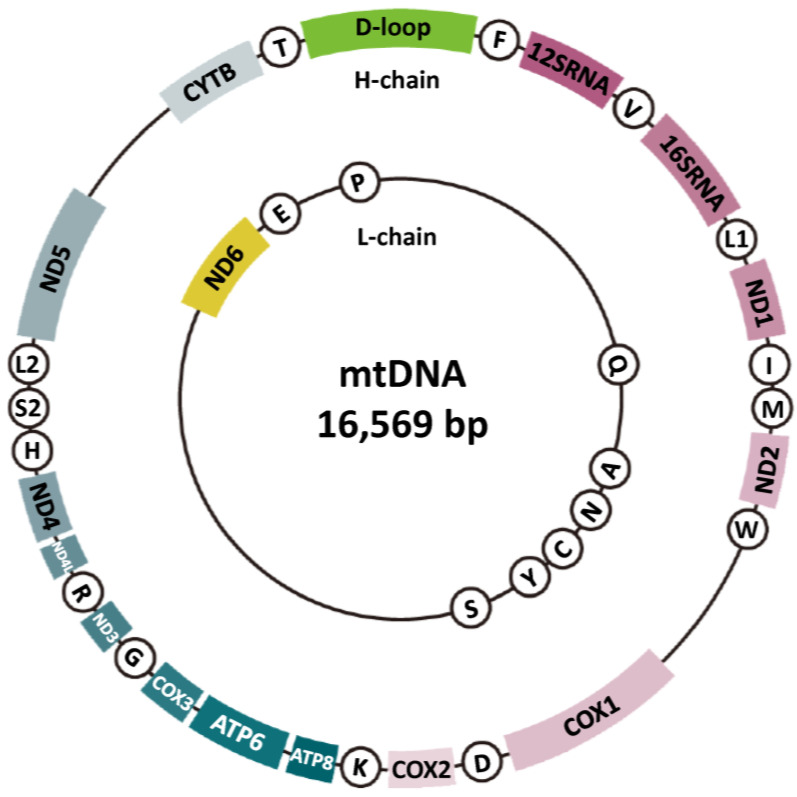
Molecular structure of mtDNA. Mitochondrial DNA (mtDNA) is a circular, double-stranded molecule located within the mitochondria, distinct from nuclear DNA. It is typically 16.5 kilobases in length and encodes 37 genes, including 13 proteins essential for oxidative phosphorylation (the region represented by the sector, excluding the D-loop), 22 transfer RNAs (tRNAs) (circled alphanumeric characters), and 2 ribosomal RNAs (12SRNA and 16SRNA) necessary for mitochondrial protein synthesis. Functionally, mtDNA plays a crucial role in energy production by facilitating the generation of adenosine triphosphate (ATP) through the electron transport chain (ETC) and maintaining mitochondrial integrity and functionality.

**Figure 3 ijms-26-01172-f003:**
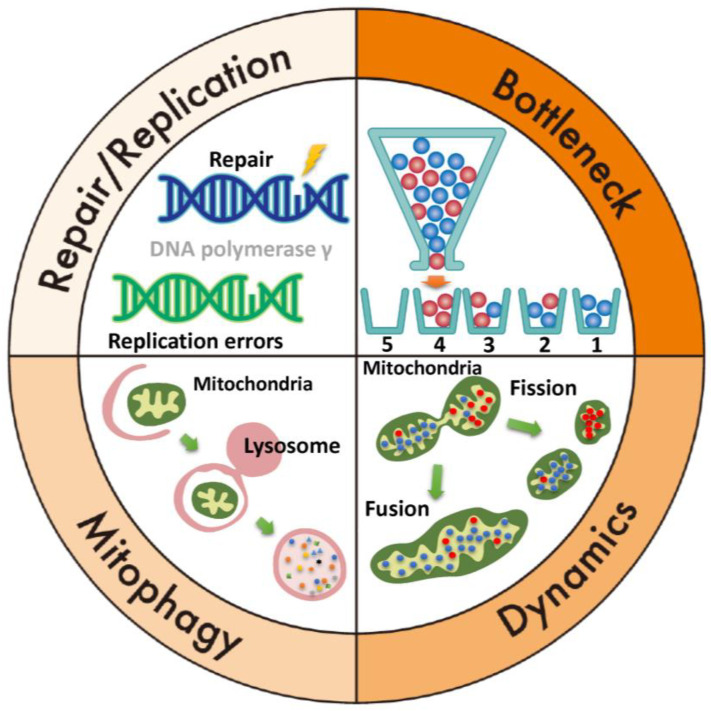
Mechanisms for maintaining mitochondrial genome stability. Top right (bottleneck theory): a bottle contains wild-type mtDNA (blue) and mutant mtDNA (red). If the mtDNA is poured sequentially from a bottle with a narrow neck, 1 and 4 will be homoplasmic, and 2 and 3 will be heteroplasmic. Bottom right: mitochondrial dynamics, which involve the processes of fusion and fission, play a critical role in maintaining mitochondrial function. Fusion allows mitochondria to mix their contents, including mtDNA, proteins, and metabolites, which helps dilute damaged mtDNA and compensate for functional defects. Fission, on the other hand, segregates damaged or dysfunctional mtDNA. Together, these dynamic processes ensure the quality control, distribution, and adaptation of mitochondria to meet cellular energy demands, ultimately restoring and preserving mitochondrial function. Bottom left (mitophagy): the damaged mitochondria are engulfed by autophagosomes and delivered to lysosomes, where they are broken down and recycled, preventing the accumulation of defective mitochondria and maintaining mitochondrial quality and function. Top left: mtDNA repair and replication are crucial for maintaining mitochondrial function and integrity. mtDNA replication is semi-autonomous and occurs within the mitochondria. It involves the replication of the circular mtDNA molecule by a set of mitochondrial enzymes, including DNA polymerase γ.

**Table 1 ijms-26-01172-t001:** A summary of mitochondrial DNA copy number data compiled from published studies to date.

Publication Year	Article Type	The Number of Women Who Participated in the Study		Mitochondrial DNA Copy Numbers in Women with PCOS	Summary	Ref.
		PCOS Patients	Controls			
2021	Original	50	60	↓	In patients with PCOS, mitochondrial DNA (mtDNA) copy numbers exhibited a negative correlation with insulin resistance, waist circumference, and triglyceride levels while showing a positive correlation with sex-hormone-binding globulin levels.	[129]
2017	Original	80	50	↓	Mutations in mitochondrial tRNAs disrupt their secondary structure, potentially leading to elevated reactive oxygen species (ROS) levels.	[110]
2019	Original	118	114	↓	PCOS cases harboring the D310 and 189G alleles may be associated with reduced mtDNA copy numbers and an elevated luteinizing hormone/follicle-stimulating hormone (LH/FSH) ratio.	[113]
2019	Original	70	59	↓	Mitochondrial tRNA mutations were suspected as causative factors in PCOS, with six cases (60%) exhibiting transition mutations and four cases (40%) displaying translocation mutations.	[114]
2020	Original	30	30	↓	A study involving 30 women with PCOS identified transition mutations in 82.35% of cases and transversion mutations in 17.64%.	[77]
2021	Original	263	326	↑	The mtDNA copy number in PCOS patients was significantly higher than that observed in the control group.	[80]
2022	Review			↓	Of 13,812 studies identified, 15 were deemed eligible for inclusion, and 8 were suitable for meta-analysis.	[5]
2023	Original	168	83	↑	The study also identified mtDNA mutations positively correlated with LH/FSH levels, which may play a protective role in the development of PCOS.	[94]
2023	Original	45	45	↓	mtDNA mutations were detected in the A6, A8, COX1, COX2, COX3, CYTB, ND1, ND2, NS3, ND4, ND5, and ND6 regions.	[84]
2023	Original	39	68	↓	Levels of malondialdehyde and 8-oxodeoxyguanosine (8-OHdG) in the follicular fluid of PCOS patients were higher than those in controls.	[130]
2024	Original	70	50	↓	Low-heterogeneity mtDNA mutations are associated with PCOS-related traits, suggesting their potential impact on both the structural flexibility and overall stability of proteins.	[106]

## Data Availability

No new data were created.

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
