# Peer review of "A Comprehensive Review of the Contribution of Mitochondrial DNA Mutations and Dysfunction in Polycystic Ovary Syndrome, Supported by Secondary Database Analysis"

_ijms, 2025, doi:10.3390/ijms26031172_

Round 1
Reviewer 1 Report
Comments and Suggestions for Authors
This narrative review focuses on the interplay between oxidative stress, mtDNA mutations and mitochondrial dysfunction in the pathophysiology of PCOS. The review is timely, due to the recent emergence of the knowledge of these mechanisms in PCOS, and exhaustive. The length and excessive detail of certain sections attempt against its readability. On the other hand, the absence of the information (provided as supplemental material) in the main text may lead the reader to feel that there is missing information.
My suggestions are:
1. Include the search strategy and inclusion criteria in the main text.
2. Shorten the main text whenever possible (to about 12-13 pages instead of 17).
3. Clarify certain sections:
a. The explanation on the impact of androgen excess (subtitle 2.1) on mitochondria is not easy to understand. The authors mention 5 mechanisms, but it is not easy to follow the link between them, or the impact on mitochondrial function and PCOS (just as an example: “Fifthly, androgens directly or indirectly activate the phosphatidylinositol 3-kinase (PI3K)/protein kinase B (PKB/AKT) signaling pathway and mTOR, supporting anabolic processes and promoting cell survival in skeletal muscle”). Please reformulate the whole section for the sake of clarity.
b. Other paragraphs that would benefit from further clarity are those on age-related frequency and distribution of mtDNA mutations, and its relationship with oxidative stress (lines 414-481). From what I understand, these long paragraphs conclude that mtDNA mutations are due to replication errors rather than oxidative stress. Thes paragraphs are too long -thus risking to deviate the reader’s attention- to finally indicate that there is little relationship between mtDNA mutations and oxidative stress. Could the idea be conveyed more simply?
Author Response
Manuscript ID: ijms-3429620
Type of manuscript: Review
Title: The intricate interplay among oxidative stress, mitochondrial dysfunction, and mitochondrial DNA mutations in the pathogenesis of polycystic ovary syndrome
Authors: Hiroshi Kobayashi, Sho Matsubara, Chiharu Yoshimoto, Hiroshi Shigetomi, Shogo Imanaka
Dear Editor in Chief:
First of all, thank you and the reviewers for the thoughtful comments and helpful suggestions on our manuscript. We have carefully considered each of the comments, made every effort to address the concerns raised, and applied corresponding revisions to the manuscript. This paper has been shortened to two-thirds of its original length at the request of reviewer 1, so it is very difficult to comment on what has been revised and how. The major changes are highlighted in blue. The detailed, point-by-point responses to the reviewer comments are given below.
We believe that our manuscript has been considerably improved as a result of these revisions, and hope that the revised manuscript is acceptable for publication in IJMS.
I would like to thank you once again for your consideration of our work and inviting me to submit the revised manuscript. I look forward to hearing from you.
With best regards,
Hiroshi Kobayashi, M.D., Ph.D.
hirokoba@naramed-u.ac.jp
Answer to the reviewer 1
This narrative review focuses on the interplay between oxidative stress, mtDNA mutations and mitochondrial dysfunction in the pathophysiology of PCOS. The review is timely, due to the recent emergence of the knowledge of these mechanisms in PCOS, and exhaustive. The length and excessive detail of certain sections attempt against its readability. On the other hand, the absence of the information (provided as supplemental material) in the main text may lead the reader to feel that there is missing information.
My suggestions are:
Comment 1:
Include the search strategy and inclusion criteria in the main text.
Response 1:
As this article is a narrative review rather than a systematic review, the Materials and Methods section has been omitted. Sorry. The search strategy and inclusion criteria have been incorporated into Section 10.
Comment 2:
Shorten the main text whenever possible (to about 12-13 pages instead of 17).
Response 2:
This article has been condensed to 14 pages; however, further reduction would prove challenging. Due to the extensive revision of the text, it is not feasible to enumerate all the modifications implemented.
Comment 3:
Clarify certain sections:
Comment 3-a:
The explanation on the impact of androgen excess (subtitle 2.1) on mitochondria is not easy to understand. The authors mention 5 mechanisms, but it is not easy to follow the link between them, or the impact on mitochondrial function and PCOS (just as an example: “Fifthly, androgens directly or indirectly activate the phosphatidylinositol 3-kinase (PI3K)/protein kinase B (PKB/AKT) signaling pathway and mTOR, supporting anabolic processes and promoting cell survival in skeletal muscle”). Please reformulate the whole section for the sake of clarity.
Response 3-a:
In section 2.1, the five mechanisms of action of androgens are simplified and discussed in relation to PCOS pathogenesis.
Comment 3-b:
Other paragraphs that would benefit from further clarity are those on age-related frequency and distribution of mtDNA mutations, and its relationship with oxidative stress (lines 414-481). From what I understand, these long paragraphs conclude that mtDNA mutations are due to replication errors rather than oxidative stress. Thes paragraphs are too long -thus risking to deviate the reader’s attention- to finally indicate that there is little relationship between mtDNA mutations and oxidative stress. Could the idea be conveyed more simply?
Response 3-b:
It seems that we provided excessive detail in an attempt to facilitate researchers' understanding of mitochondrial DNA structure (the presence of heavy and light chains), strand bias in DNA mutations, and the mechanisms underlying base mutations in this paper. Reviewer 2 also noted that the descriptions resembled textbook explanations. We have endeavored to present the content of Section 5 as concisely as possible and kindly request a further peer review.
Reviewer 2 Report
Comments and Suggestions for Authors
Subject: Major Revision Feedback on Manuscript: "The Intricate Interplay Among Oxidative Stress, Mitochondrial Dysfunction, and Mitochondrial DNA Mutations in the Pathogenesis of Polycystic Ovary Syndrome"
Dear Authors,
Thank you for the opportunity to review your manuscript. The topic, linking mitochondrial mutations and dysfunction with PCOS, is both timely and relevant, given that PCOS pathogenesis remains incompletely understood. Your work has the potential to contribute valuable insights to the field. However, I believe the manuscript requires major revisions to clarify the study design, methodology, and overall research goals. Below are my specific comments:
- Title
The current title should more explicitly reflect the study’s design and main objective. Since the abstract suggests this is a secondary analysis of a public database focusing on mitochondrial mutations in PCOS, consider revising the title to explicitly mention the methodology (e.g., “A Secondary Database Analysis of …”) and the primary aim (e.g., “… Linking Mitochondrial Mutations to PCOS Pathogenesis”). - Abstract and Objectives
From the abstract, it appears you performed a secondary analysis of publicly available data. This information is crucial and should be stated more clearly. Explicitly articulate how you intend to link mitochondrial DNA mutations and dysfunction to PCOS. Clear objectives will help readers understand the scope and significance of your research. - Introduction
A stronger and more detailed introduction to PCOS is needed, particularly regarding the two main pathophysiological pathways, excess androgen production and insulin resistance, and how these can manifest independently or in overlapping clinical spectrums. By highlighting these established mechanisms, you can then better position the potential role of mitochondrial dysfunction within this complex condition. - Materials and Methods
The absence of a dedicated Materials and Methods section (or a clear methodological outline) makes it challenging to follow your study’s rationale and interpret your findings. Please provide: - A clear description of your data source(s) and inclusion/exclusion criteria.
- Detailed methods used to analyze mitochondrial DNA mutations.
- Any statistical approaches applied, including software and significance thresholds.
- Results and Discussion
Without the methodological details, the logical flow of the results is difficult to assess. As presented, from Chapter 2 on, reads more like a book chapter than a systematic analysis. Reorganize the content to focus on the key results from your analysis, followed by a clear discussion linking these findings to existing literature and to the broader understanding of PCOS etiology. - Overall Assessment
Your research question is compelling, and the topic holds promise for advancing our understanding of PCOS. However, in its current form, the manuscript would benefit from major revisions to: - Clearly define the study’s aim and design.
- Provide a full methodological description.
- Strengthen the introduction with a more comprehensive overview of PCOS pathogenesis.
- Present a cohesive, method-driven results and discussion section.
Best regards
Author Response
Manuscript ID: ijms-3429620
Type of manuscript: Review
Title: The intricate interplay among oxidative stress, mitochondrial dysfunction, and mitochondrial DNA mutations in the pathogenesis of polycystic ovary syndrome
Authors: Hiroshi Kobayashi, Sho Matsubara, Chiharu Yoshimoto, Hiroshi Shigetomi, Shogo Imanaka
Dear Editor in Chief:
First of all, thank you and the reviewers for the thoughtful comments and helpful suggestions on our manuscript. We have carefully considered each of the comments, made every effort to address the concerns raised, and applied corresponding revisions to the manuscript. This paper has been shortened to two-thirds of its original length at the request of reviewer 1, so it is very difficult to comment on what has been revised and how. The major changes are highlighted in blue. The detailed, point-by-point responses to the reviewer comments are given below.
We believe that our manuscript has been considerably improved as a result of these revisions, and hope that the revised manuscript is acceptable for publication in IJMS.
I would like to thank you once again for your consideration of our work and inviting me to submit the revised manuscript. I look forward to hearing from you.
With best regards,
Hiroshi Kobayashi, M.D., Ph.D.
hirokoba@naramed-u.ac.jp
Answer to the reviewer 2
Subject: Major Revision Feedback on Manuscript: "The Intricate Interplay Among Oxidative Stress, Mitochondrial Dysfunction, and Mitochondrial DNA Mutations in the Pathogenesis of Polycystic Ovary Syndrome"
Dear Authors,
Thank you for the opportunity to review your manuscript. The topic, linking mitochondrial mutations and dysfunction with PCOS, is both timely and relevant, given that PCOS pathogenesis remains incompletely understood. Your work has the potential to contribute valuable insights to the field. However, I believe the manuscript requires major revisions to clarify the study design, methodology, and overall research goals. Below are my specific comments:
Comment 1:
Title
The current title should more explicitly reflect the study’s design and main objective. Since the abstract suggests this is a secondary analysis of a public database focusing on mitochondrial mutations in PCOS, consider revising the title to explicitly mention the methodology (e.g., “A Secondary Database Analysis of …”) and the primary aim (e.g., “… Linking Mitochondrial Mutations to PCOS Pathogenesis”).
Response 1:
Oxidative stress resulting from androgen excess and insulin resistance has been identified as a contributing factor in the pathogenesis of PCOS, with evidence suggesting that oxidative stress induces mitochondrial dysfunction. Recent advancements in mtDNA analysis techniques have led to the identification of numerous rare mtDNA mutations, though it has been concluded that transversion mutations, which are characteristic of oxidative stress, remain infrequent. This narrative review was undertaken to address this issue. In this article, data from prior studies were systematically compiled in Excel (Supplementary Table S1), and the proportions of transition and transversion mutations were recalculated. Consequently, no specialized analytical techniques or statistical procedures were employed. As such, this manuscript does not primarily represent a secondary analysis of publicly available databases. We considered alternative titles, such as "Reconsideration of the Pathogenesis of PCOS Based on Secondary Database Analysis of Mitochondrial DNA Mutations" or "Rethinking the Pathogenesis of PCOS through Database Analysis of Mitochondrial DNA Mutations," but there was concern that these titles might suggest the paper is primarily a secondary analysis. Therefore, we prefer to retain the original title.
Comment 2:
Abstract and Objectives
From the abstract, it appears you performed a secondary analysis of publicly available data. This information is crucial and should be stated more clearly. Explicitly articulate how you intend to link mitochondrial DNA mutations and dysfunction to PCOS. Clear objectives will help readers understand the scope and significance of your research.
Response 2:
We added the following text in the abstract.
We are rethinking the pathogenesis of PCOS based on these database analyses.
Comment 3:
Introduction
A stronger and more detailed introduction to PCOS is needed, particularly regarding the two main pathophysiological pathways, excess androgen production and insulin resistance, and how these can manifest independently or in overlapping clinical spectrums. By highlighting these established mechanisms, you can then better position the potential role of mitochondrial dysfunction within this complex condition.
Response 3:
This article has been condensed to 14 pages as per reviewer 1's instructions.
At the end of the first paragraph, we added the following:
The pathophysiology primarily involves two mechanisms: excessive androgen production and insulin resistance, which independently or interactively contribute to the clinical manifestations, including reproductive, dermatologic, and metabolic features. Hyperandrogenism is central to the disorder, often resulting from dysregulation of the hypothalamic-pituitary-ovarian axis or ovarian dysfunction [11-15]. Additionally, insulin resistance, a hallmark feature of PCOS, substantially contributes to its metabolic and reproductive manifestations and is often accompanied by compensatory hyperinsulinemia [16]. Extensive studies have examined the molecular interplay between androgen excess and insulin resistance [17].
Additionally, at the end of the second paragraph, we added the following:
While substantial evidence indicates that mitochondrial dysfunction driven by oxidative stress plays a pivotal role in the pathogenesis of PCOS, investigations into mtDNA mutations have not yet yielded conclusive evidence supporting this link. Consequently, we undertook an effort to compile and analyze the existing data on mtDNA mutations reported in connection with PCOS. Understanding the factors contributing to mtDNA mutations and deletions, as well as their repair mechanisms, is essential for elucidating the pathophysiology of PCOS and translating these findings into therapeutic strategies.
Comment 4:
Materials and Methods
The absence of a dedicated Materials and Methods section (or a clear methodological outline) makes it challenging to follow your study’s rationale and interpret your findings. Please provide:
A clear description of your data source(s) and inclusion/exclusion criteria.
Detailed methods used to analyze mitochondrial DNA mutations.
Any statistical approaches applied, including software and significance thresholds.
Response 4:
As this article is a narrative review rather than a systematic review, the Materials and Methods section has been omitted. Sorry. The search strategy and inclusion criteria have been incorporated into Section 10.
In this article, data from prior studies [5,48,77,84,94,106,109,110,112-114] were systematically compiled in Excel (Supplementary Table S1), and the proportions of transition and transversion mutations were recalculated. Consequently, no specialized analytical techniques or statistical procedures were employed. As such, this manuscript does not primarily represent a secondary analysis of publicly available databases.
Comment 5:
Results and Discussion
Without the methodological details, the logical flow of the results is difficult to assess. As presented, from Chapter 2 on, reads more like a book chapter than a systematic analysis. Reorganize the content to focus on the key results from your analysis, followed by a clear discussion linking these findings to existing literature and to the broader understanding of PCOS etiology.
Response 5:
This article has been condensed to 14 pages as per reviewer 1's instructions.
We have endeavored to reorganize the content to focus on key issues in each section and summarize our findings.
Comment 6:
Overall Assessment
Your research question is compelling, and the topic holds promise for advancing our understanding of PCOS. However, in its current form, the manuscript would benefit from major revisions to:
Clearly define the study’s aim and design.
Provide a full methodological description.
Strengthen the introduction with a more comprehensive overview of PCOS pathogenesis.
Present a cohesive, method-driven results and discussion section.
Response 6:
Based on your feedback, we have shortened the article by about two-thirds and made some major revisions, so we would be grateful if you could re-review it.